# Internal Morphology and Phylogenetic Position of *Mycetomorpha vancouverensis* (Pancrustacea: Rhizocephala), an Enigmatic Parasitic Barnacle

**DOI:** 10.3390/biology13120968

**Published:** 2024-11-24

**Authors:** Aleksei Miroliubov, Anastasia Lianguzova, Darya Krupenko, Liudmila Poliushkevich, Semyon Novokreshchennykh, Natalia Arbuzova, Georgii Kremnev

**Affiliations:** 1Laboratory of Parasitic Worms and Protists, Zoological Institute RAS, Universitetskaya Embankment 1, Saint-Petersburg 199034, Russia; anastasialianguzova@gmail.com (A.L.); krupenko.d@gmail.com (D.K.); arbuzovanata0211@gmail.com (N.A.); glistoved69@gmail.com (G.K.); 2Department of Invertebrate Zoology, Saint-Petersburg State University, Universitetskaya Emb., 7/9, Saint-Petersburg 199034, Russia; 3Department of Embryology, Saint-Petersburg State University, Universitetskaya Emb., 7/9, Saint-Petersburg 199034, Russia; mila.papurika@gmail.com; 4Sakhalin Branch of the Russian Federal Research Institute of Fisheries and Oceanography, Komsomolskaya Street, 196, Yuzhno-Sakhalinsk 693023, Russia; novokreshennihsv@sakhniro.vniro.ru

**Keywords:** parasitic barnacles, Mycetomorphidae, phylogeny, interna, invasive rootlets, host–parasite interaction

## Abstract

*Mycetomorpha vancouverensis* is an enigmatic parasitic barnacle from the family Mycetomorphidae. This species, like other Rhizocephala, undergoes significant morphological transformation due to its parasitic lifestyle. The adult females have an externa, a reproductive organ, and an interna, a system of rootlets that infiltrate the host for feeding and host manipulation. The phylogenetic position of *M. vancouverensis* has been unclear for a long time as it shares characteristics of both basal and diverged families. The molecular analysis of the 18S rDNA confirmed its close relationship with the representatives of the family Peltogastridae, helping to place it on the evolutionary tree. In addition, we provided a detailed description of the morphology and histology of both the externa and interna. The interna branches out into trophic and invasive rootlets, with the latter invading the host’s nervous system. These invasive rootlets are organized into goblet-shaped organs, which are present in basal groups. Thus, our morphological data support the hypothesis that *Mycetomorpha* represents an early evolutionary branch within Rhizocephala.

## 1. Introduction

Parasitic barnacles, the Rhizocephala, are among the most impressively modified metazoan parasites. Adults of the rhizocephalans are dramatically transformed due to their endoparasitic lifestyle and do not resemble their free-living relatives [1]. The body of a reproducing female consists of two functional parts: an interna, a system of trophic rootlets infiltrating the host’s body, and an externa, a temporary reproductive organ [2,3]. Males are extremely reduced and incorporated into the externa [2,3]. 

The rhizocephalans are capable of manipulating their hosts and taking control over them. Parasitic barnacles alter the metabolism, morphology, and even behavior of infected specimens. The most striking example is the feminization of male crabs [4,5,6,7,8,9,10]. Besides its trophic function, the interna is responsible for host manipulation [11,12,13,14]. The interna has specialized invasive rootlets that invade the host’s nervous ganglia. The ultrastructure of such invasive rootlets differs from the one of the common trophic rootlet and suggests high synthetic activity [11,12,13,14].

Invasive rootlets were found in all of the examined rhizocephalan species [11,12,13,14,15,16]. These rootlets can be classified into two types: goblet-shaped organs and neuropil rootlets. Goblet-shaped organs are typically found in the peripheral areas of the ganglia, primarily within the neuronal somata layer, and have modified distal ends [11,12]. On the contrary, neuropil rootlets do not exhibit significantly modified distal tips, and they protrude deeper into the ganglion neuropil with changing structure along the rootlet [14]. The type of invasive rootlets that a species possesses correlates with its phylogenetic position inside the Rhizocephala. Representatives of more basal families have goblet-shaped organs, while in the derived taxa, these organs are totally replaced by neuropil rootlets [13,14]. The structure of invasive rootlets is unknown in many families of the Rhizocephala. Because these invasive rootlets play an important role in regulating host–parasite interactions [14], studying their morphology is crucial for understanding the evolution of the whole group of the parasitic barnacles.

The family Mycetomorphidae is one of the most mysterious and unexplored among rhizocephalans. It comprises only three known species distributed in the Northern Pacific: *Mycetomorpha vancouverensis* Potts, 1912; *M. albatrossi* Høeg and Rybakov, 1996; and *M. abyssalis* Kakui, 2024. *M*. *vancouverensis* was first described from the coastal waters of Vancouver by Potts, 1912 [17]. Later, it was recorded from the inshore waters of Alaska, Kuril Islands, and Sakhalin [18]. Another species, *M. albatrossi*, was found in the collections of Potts from British Columbia [18]. Recently, one more species, *M. abyssalis*, was described from the deep sea near the Japanese coastline [19]. 

The phylogenetic position of the family Mycetomorphidae is still uncertain. Previously, rhizocephalans were split into two suborders, Kentrogonida and Akentrogonida, based on the life cycle and morphological features [2,20,21]. Mycetomorphidae was placed among other akentrogonids in the first phylogenetic analysis conducted on cumulative characteristics such as morphology of externa and cyprid larvae and presence/absence of male cyprid instar [22]. However, the mycetomorphid cyprid larvae are similar to the kentrogonids [18]. 

According to the molecular data, suborders Kentrogonida and Akentrogonida are not valid anymore [23,24], both appeared polyphyletic. Most of the “akentrogonids” form long branches within the phylogenetic tree. The Mycetomorphidae is not related to them; it was resolved to be among basal-branched taxa, close to the genera *Peltogaster* and *Lernaeodiscus* [23], both belonging to the family Peltogastridae in the present classification. Further molecular studies involved more peltogastrid species but did not take the Mycetomorphidae into the dataset [24]. Thus, its position remains uncertain. Revising molecular data on the Mycetomorphidae, together with new data on its structure of the interna and the invasive rootlets, may help to resolve the phylogeny of this family.

Here, we aimed to describe the gross morphology and histological organization of externa and interna in *M. vancouverensis* collected near the South Kuril Islands. Additionally, the species identity was checked by the comparison of 18S rDNA sequences with the previous molecular data from Høeg et al. (2019) [23]. Finally, based on the 18S rDNA dataset embracing available “kentrogonid” species, we provide a new reconstruction of the Mycetomorphidae phylogeny.

## 2. Materials and Methods

### 2.1. Sampling

Sampling was performed using a bottom trawl during the expedition of the research vessel “Dmitriy Peskov” in October–November 2019, organized by the Russian Federal Research Institute of Fisheries and Oceanography in the coastal waters of the South Kuril Islands, occurring near Kunashir Island at three localities close to each other (43°36′4″ N, 146°10′3″ E; 44°01′2″ N, 145°55′2″ E; 43°50′7″ N, 146°05′2″ E) (Figure 1A) from depths of 20–54 m. Three obtained shrimp specimens were infected by rhizocephalan that were morphologically similar to *Mycetomorpha vancouverensis*. They were fixed in 96% ethanol. The host was identified through morphology as *Neocrangon communis* (Rathbun, 1899) (Figure 1B). One specimen of infected shrimp was deposited in the collection of parasitic crustaceans at the Zoological Institute of the Russian Academy of Sciences (ZIN RAS).

Images of the externae were taken using the M125 C (Leica, Wetzlar, Germany) stereomicroscope equipped with a Flexicam C3 camera (Leica, Wetzlar, Germany).

### 2.2. Molecular Analysis

A piece of rhizocephalan’s externa and a fragment of the host’s pleonal muscles were used to extract DNA. They were taken from 96% ethanol and dried completely at 35 °C. Next, specimens were incubated in 200 μL of 5% solution of Chelex^®^ 100 resin (Bio-Rad, Hercules, CA, USA) with 0.2 mg mL−1 proteinase K (Evrogen, Moscow, Russia) at 56 °C for 6 h, then kept for 8 min at 90 °C and centrifuged for 10 min at 16,000× *g*. The supernatant containing DNA was transferred to a new tube and stored at −20 °C.

Fragments of 18S rDNA were amplified with three pairs of primers: 18Sf (5′-TACCTGGTTGATCCTGCCAG-3′) and 614r (5′-TCCAACTACGAGCTTTTTAACC-3′) for fragment 1, 554f (5′-AAGTCTGGTGCCAGCAGCCGC-3′) and 1282r (5′-TCACTCCACCAACTAAGAACGGC-3′) for fragment 2, and 1150f (5′-ATTGACGGAAGGGCACCACCAG-3′) and 18Sr (5′-TAATGATCCTTCCGCAGGTTCAC-3′) for fragment 3 [25]. The 20 μL reaction mixture contained 4 μL of ScreenMix-HS (Evrogen), 0.5 μL of each primer (10 pmol/μL), 2 μL of the DNA, and 13 μL of PCR-grade water (Evrogen). Polymerase chain reactions (PCRs) were run on a BioRad T100 thermal cycler (Bio-rad Laboratories Inc., Hercules, CA, USA) with the following thermocycling profile: 5 min denaturation at 95 °C, followed by 35 cycles of: 1 min at 94 °C; 1 min at 55 °C (fragment 1), or 59 °C (fragment 2), or 57 °C (fragment 3); and 2 min at 72 °C, with a final 10 min extension at 72 °C. PCR products were stained with ethidium bromide (0.5 μg/mL) and visualized through electrophoresis on a 1% agarose gel. Sequencing of the target PCR products was further performed with PCR primers on an AB3500xL genetic analyzer (Applied Biosystems, Woburn, MA, USA). Chromatograms were analyzed and aligned in Geneious Prime (https://www.geneious.com accessed on 20 October 2024). A newly generated sequence of 18S rDNA of *Mycetomorpha* was submitted to GenBank under accession number PQ408617, and the 18S rDNA sequence of the host, *N. communis*, was submitted under accession number PQ408618. The annotated sequence of the ribosomal operon of *Parasacculina yatsui* (Boschma, 1936) (MG604305, [26]) was used to map the *M. vancouverensis* sequence. 

For the phylogeny reconstruction, we reviewed all available 18S rDNA sequences of Rhizocephala from GenBank. Provisional analyses showed that the species of “Akentrogonida” form long branches and thus negatively affect support values in the whole tree. Thus, we removed “akentrogonids” from the final dataset (Table 1). Thoracicans were added to represent the outgroup. The substitution model for the Bayesian inference (BI) analysis was selected in bModelTest [27]; it was Tim+G+I. The BI analysis was conducted using the Monte Carlo Markov Chain (MCMC) analysis available in Bayesian Evolutionary Analysis by Sampling Trees (BEAST2) [28] on XSEDE at the CIPRES Science Gateway (https://www.phylo.org accessed on 20 October 2024). Three independent runs of MCMC were performed, each with 10,000,000 generations and sampling every 1000 generations. The trace files were analyzed with Tracer v1.7 [29]. The log files were combined using LogCombiner, discarding the first 10% as burn-in. Trees were summarized with TreeAnnotator using the maximum clade credibility tree option and with node heights as mean heights. The maximum likelihood (ML) analysis was run in IQ-TREE 1.6.12 [30] with UFBoot2 (ultrafast bootstrap) [31] with 1000 replicates. According to the results of test runs, the best model for ML analysis was TN93+G+I, the one estimated by MEGA11 [32].

### 2.3. Histology

The shrimps were dissected in water under a MBS 10 LOMO stereomicroscope (Moscow, Russia) to isolate parts of the interna and host’s ventral nerve cord. Then, specimens were dehydrated in a graded alcohol series and embedded in Histomix™ medium (BioVitrum, St Petersburg, Russia). Sections were cut at 5 μm using a Minux S700 RWD (Shenzhen, China) microtome, stained with Mallory’s trichrome stain, dehydrated in alcohol series, cleared in xylene, and mounted in BioMount medium (Bio Optica, Milano, Italy). The slides were examined under a Leica DM2500 microscope (Leica Microsystems, Wetzlar, Germany), and photos were taken using a Nikon DS-Fi1 camera (Nikon, Tokyo, Japan).

### 2.4. Scanning Electron Microscopy (SEM)

The specimens for SEM were dehydrated in an ethanol series and acetone, then dried in a critical point dryer Autosamdri-815, Series A (Tousimis, Rockville, MD, USA). They were mounted on stubs, coated with platinum using a Giko IB-5 ion coater, and viewed under an Quanta 250 (FEI, Hillsboro, Oregon, USA) scanning electron microscope at the “Taxon” Research Resource Center of ZIN RAS.

## 3. Results

### 3.1. Molecular Analysis

We obtained an almost complete 18S rDNA sequence (1816 bp) from one of our specimens. Through the BLAST search, it was found to be identical with a GenBank sequence of *M. vancouverensis* (MH974514 from [23]) in the overlapping part (1757 bp). The percent identity with *M. abyssalis* (LC799152 from [19]) was 99.34%. Other rhizocephalan sequences of 18S rDNA demonstrated 97.46% and less identity with our sequence.

The newly generated sequence was used to resolve the phylogenetic position of the Mycetomorphidae. An alignment was built with 39 sequences of “kentrogonid” rhizocephalans and three species of Thoracica as the outgroup (Table 1). After trimming, the alignment was 1977 bp long, including gaps. Both BI and ML trees supported sister relationships between two species of *Mycetomorpha* (Figure 2). In the Bayesian tree, the mycetomorphids were resolved with high support as a sister group with various species of the Peltogastridae: *Briarosaccus* spp., *Peltogaster* spp., *Galatheascus striatus*, and *Tortugaster boschmai*. The ML analysis confirmed the common root of this group; however, the interrelationships within it were different: the Mycetomorphidae formed a common branch with *Peltogaster lineatus* and *Briarosaccus hoegi*, with moderate support. In both BI and ML analyses, the family Peltogastridae appeared polyphyletic. *Paratriangulus galatheae* and *Septosaccus rodriguezii* formed separate basal branches within a large clade comprising the Polyascidae, Sacculinidae, Parthenopeidae, and Peltogasterellidae. The representatives of the peltogastrid genus *Lernaeodiscus* grouped with *Cyphosaccus norvegicus* (Peltogasterellidae), and in the BI tree, this branch grouped with the Polyascidae and Sacculinidae. The Peltogasterellidae were also resolved as polyphyletic in our trees, as *Peltogasterella sulcata* grouped with the Sacculinidae with high support. Monophyly of the families Sacculinidae and Polyascidae was well supported.

### 3.2. Externa Morphology

The parasite’s externa is located on the ventral side of the host’s pleonal segments and is connected to the interna via a stalk. The interna is not extensive and is located near the ventral surface of the pleon segments, with the main trunk running along the host’s ventral nerve cord. In the host’s pereon, the main trunk branches out into a system of trophic rootlets that are randomly positioned between the hepatopancreas tubules (Figure 3).

The externa has an elongated shape in the anterior–posterior direction, with multiple tubular lobes protruding from its central part (Figure 4A). The stalk is positioned at the center of the surface facing the host pleon (further is the “ventral” surface, the opposite surface is the “dorsal” surface) and is surrounded by a special shield of thick cuticle (Figure 4B). The cuticle covering the externa is smooth and lacks any projections (Figure 4C,D).

The cuticle covering the externa tubular lobes is about 100–150 μm. It consists of a thick inner layer and a thin outer layer (Figure 5A,B). Beneath the cuticle, there is a hypodermal layer with circular muscular fibers under it (Figure 5A,B,D). The inner mantle cuticle is thin and smooth (Figure 5E). The mantle cavity is filled with numerous cyprid larvae (Figure 5C).

The body wall in the central part of the externa is similar to that observed in the tubular lobes (Figure 6A). There are numerous spermatogenic bodies in the mantle cavity of the externa’s central part (Figure 6B,C).

### 3.3. Interna’s Trophic Rootlets Morphology

The main trunk of the interna runs along the host’s ventral nerve cord (Figure 3, Figure 7 and Figure 8A). The main trunk is tube-shaped with a voluminous central lumen and is covered by a thick cuticle (Figure 7 and Figure 8A,B). Underneath the cuticle, there is a hypodermal cell layer. The space between the hypodermal layer and the central lumen contains various cells and structures (Figure 8C,D). These are muscular fibers, branched cells, cells with fuchsin-stained content, and spherical structures that consist of two or three layers of cells and granular aniline-stained contents in the center (Figure 7 and Figure 8A,B). In one specimen, we unexpectedly observed spermatogenic bodies (Figure 8E,F) and cyprid larvae (Figure 8G) in the central lumen of the main trunk. 

The main trunk gives rise to side branches. The highest number of side branches is found near the ventral side of the host pleonal segments (Figure 3). These rootlets have a specific histological structure. The outer surface is covered by a thin cuticle covering the hypodermal layer, followed by the axial layer. The center of the rootlet contains a lumen which may be filled with fibrous contents in some areas (Figure 9A,E,F). Unusual hollow cellular structures (henceforth named as bubble-like) are occasionally present in the lumen of the interna side branches (Figure 9A–D). These bubble-like structures consist of a single layer of high cuboidal cells surrounding a large cavity in the center (Figure 9A–D).

In the host’s pereon, the main trunk divides into a system of rootlets that randomly branch out and infiltrate the hepatopancreas tubules (Figure 3). These thoracic rootlets have the same organization as the side branches in the pleon (Figure 9C,D).

### 3.4. Invasive Rootlets

Part of the side branches invade the ventral nerve cord in the host’s pleon. Invasive rootlets were found in ganglia (Figure 10A–C), with connectives between them (Figure 10D,E) and peripheral nerves (Figure 10F). Distal parts of the invasive rootlets are modified into goblet-shaped organs. The goblet-shaped organs consist of two cell layers. The outer cells are flat, and their shape resembles the hypodermal cells of the trophic rootlets (Figure 10A). In some of the goblet-shaped organs, orange G-stained contents are present in the cells of the outer layer (Figure 10B). The inner cells underlying the funnel are cuboidal (Figure 10A,D). The funnel of the goblet-shaped organ is very narrow.

## 4. Discussion

### 4.1. Molecular Analysis

The structure of externa in our specimens indicates that they belong to the species *Mycethomorpha vancouverensis*. The same is confirmed by the 18S rDNA analysis: our sequence was identical with the previously published one (MH974514 from Høeg et al. [23]). Previously, molecular analysis by Høeg et al. [23] showed that the Mycetomorphidae is close to “kentrogonids” (family Peltogastridae) rather than “akentrogonids”, as had been supposed before [22]. In the present study, we used an alternative dataset to resolve the position of the Mycetomorphidae more precisely. Our results partially comply with the ones of Høeg et al. [23]: the family Mycetomorphidae is resolved as a sister group to the representatives of the family Peltogastridae. However, an important difference is in the position of the peltogastrids themselves: they are resolved as polyphyletic. The Mycetomorphidae is close to the genera *Briarosaccus*, *Peltogaster*, *Galatheascus*, and *Tortugaster*. Meanwhile, the representatives of *Lernaeodiscus*, *Septosaccus*, and *Paratriangulus* branch separately from the main part of the Peltogastridae. It should be noted that the genera *Lernaeodiscus* and *Triangulus* had been considered as representatives of a separate family Lernaeodiscidae before [3]. This family was proposed to be polyphyletic and thus was determined invalid by Høeg et al. [24]. *Triangulus munidae* was transferred to a separate family Triangulidae; *Triangulus galatheae* was renamed into *Paratriangulus galatheae* and placed within the Peltogastridae together with *Lernaeodiscus*. Our results show that reconsideration of *Lernaeodiscus* and *Paratriangulus* position should be conducted. Two members of the family Peltogasterellidae are also unrelated in our tree, with *Peltogasterella sulcata* being close to the Sacculinidae with high support. Such differences from the previous reports [23,24] may be a result of taxa sampling in our analysis, and specifically, removing long branches, the species of “Akentrogonida”. In the earlier reconstructions, rapidly evolving “akentrogonids” could affect the general tree topology, as the presence of heterogeneous branch lengths can lead to erroneous tree inferences [44,45]. Here, we tried to avoid this problem. The disadvantage of the current reconstruction is that it relies solely on a single locus: 18S rDNA. However, this allows us to take more “kentrogonid” taxa into the dataset, because presently, there are no 16S or/and 28S data on many genera within the Peltogastridae, like *Tortugaster*, *Galatheascus*, *Septosaccus*, *Paratriangulus*, as well as on the other “kentrogonid” rhizocephalans. To support the outcomes from our molecular analysis, a wide set of taxa should be tested using multiple conserved loci in the future.

### 4.2. Externa Morphology

Externa morphology differs significantly from that of closely related families and has unique features resembling those of more derived “akentrogonids”.

Our results correspond with the previous descriptions of *M. vancouverensis* externa [17,18]. It has a complex shape with copious peripheral lobes. In the central part of the externa, there are numerous spermatogenic bodies incorporated into the externa connective tissue. Notably, such typical structures for basal rhizocephalan families as male receptacles are totally absent. This feature is common among diverged rhizocephalan taxa [18,24].

### 4.3. Interna Morphology

The interna of representatives of the family Mycetomorphidae was poorly studied in previous research. Even the description of gross morphology is absent in the literature.

Here, we have shown that the interna of *M. vancouverensis* consists of a prominent main trunk that bears numerous side branches. In the host pereon, the main trunk gives rise to a network of randomly ramifying rootlets. This type of organization resembles the interna in some basal families like the Peltogastridae [3,46,47].

However, the histological structure of the main trunk body wall is rather unusual in comparison with closely related species. In peltogastrids, the body wall of the main trunk consists of a cuticle, a prominent layer of hypodermal cells, and an underlying layer of axial cells. Muscular fibers are located between axial cells [3,46,47]. In *M. vancouverensis*, the main trunk body wall, besides “normal” hypodermal, axial cells, and muscular fibers, contains two types of unique structures. Within the axial layer, there are branched cells and multicellular bubble-like structures with unclear function. Muscular fibers are probably involved in transport functions like in other rhizocephalans [46,47,48,49].

One of the studied specimens had numerous spermatogenic bodies and cyprid larvae freely floating in the central lumen of the main trunk. Even though the parasite looked intact, we suppose that this phenomenon could be explained by a presumptive inner damage of the externa.

### 4.4. Invasive Rootlets

In *M. vancouverensis*, as in all the other examined representatives of the parasitic barnacles, we observed specialized invasive rootlets [11,12,13,14,15,16]. The invasive rootlets of *M. vancouverensis* are modified into the goblet-shaped organs with two cell layers. A similar type of organization was described for *Peltogaster paguri* (Peltogastridae) [11]. In other rhizocephalan families, invasive rootlets possess a different structure. In *Peltogasterella gracilis* (Peltogasterellidae), the wall of the goblet-shaped organs consists of a single layer of cells [11], and there are three cell layers in the goblet-shaped organs of *Sacculina pugettiae* (Sacculinidae) [12]. In the diverged Polyascidae and Thompsoniidae, goblet-shaped organs were lost; instead of them, there are numerous neuropil rootlets [13,14]. The shape of invasive rootlets is morphological evidence supporting the basal position of the Mycetomorphidae on the phylogenetic tree of the parasitic barnacles and close relatedness to the Peltogastridae. Moreover, according to our unpublished data on the morphology of interna and invasive rootlets of the genera *Lernaeodiscus* and *Septosaccus*, these representatives differ from *Peltogaster* and *Mycetomorpha*. These data support our assumption of the polyphyletic status of the family Peltogastridae.

## 5. Conclusions

The representatives of the Mycetomorphidae family have a unique combination of controversial morphological features, making it difficult to determine their exact phylogenetic position. On the one hand, the structure of externa and the absence of naupliar stages resemble “akentrogonids”, while on the other hand, the cyprid morphology is typical for “kentrogonids”. The low number of sequenced rhizocephalans has also been a cause of the unclear phylogenetic position of the family Mycetomorphidae for a long time.

Both previous and current molecular analyses have revealed the Mycetomorphidae to be close to the representatives of the Peltogastridae, thus giving rise to several significant evolutionary assumptions. Notably, certain morphological features, such as the gross morphology of the interna with the main trunk bearing side branches in the pleon and thoracic rootlets, resemble the organization of the peltogastrid interna. Moreover, the invasive rootlets discovered in *M. vancouverensis* are organized as goblet-shaped organs with two cell layers and, thus, exhibit similarities to those of the peltogastrids. However, it is noteworthy that the externa structure with spermatogenic bodies does not conform to the typical characteristics of any other “kentrogonids”. This observation proves that these “akentrogonid”-like features possibly evolved independently at least twice within the Rhizocephala. Furthermore, the histological organization of the interna is recognized as being intricate and distinctive within the rhizocephalans.

## Figures and Tables

**Figure 1 biology-13-00968-f001:**
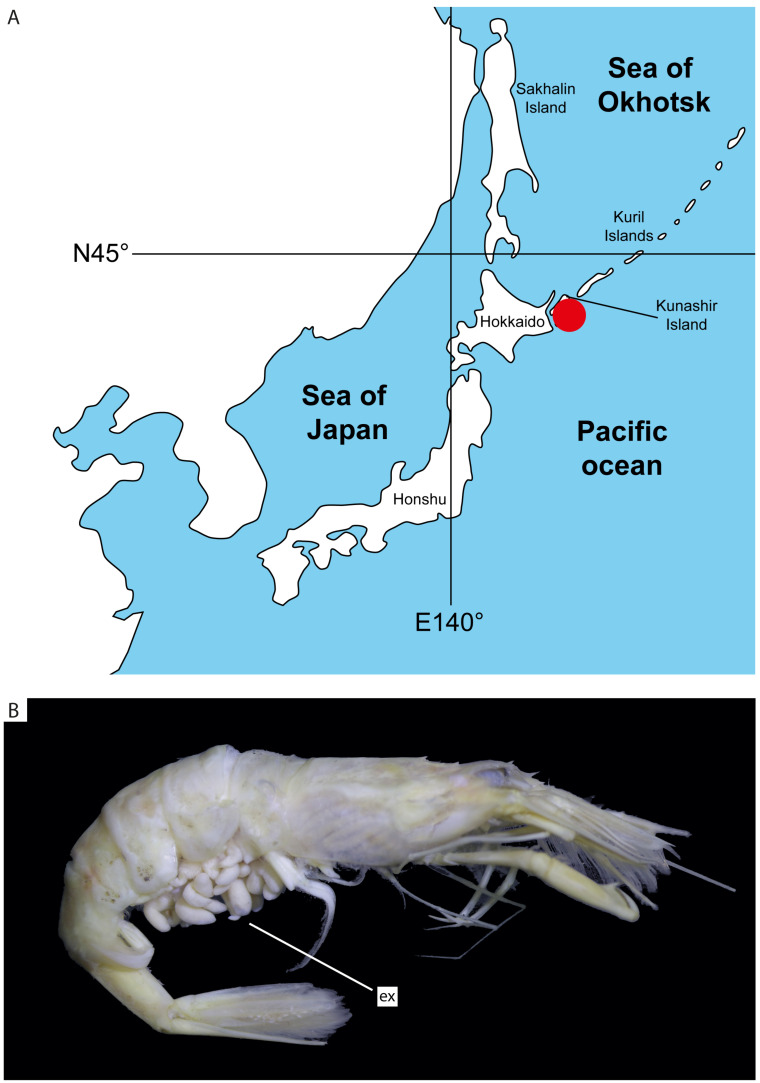
(**A**) Map of sampling site red dot indicates sampling site; (**B**) general view of *Neocrangon communis* infected by *Mycetomorpha vancouverensis*; ex, externa.

**Figure 2 biology-13-00968-f002:**
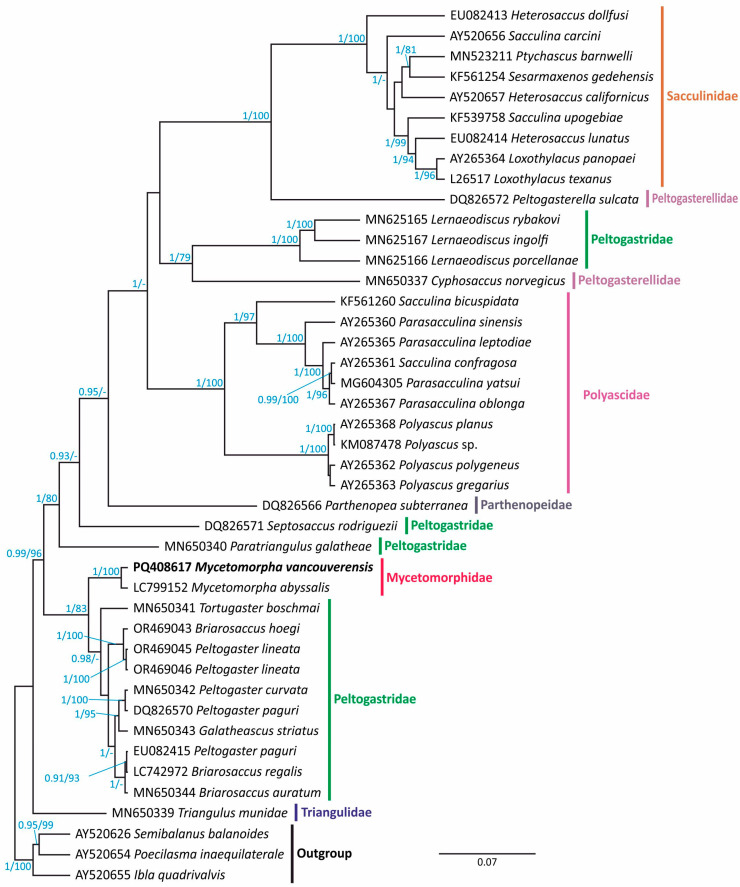
Bayesian phylogenetic tree inferred from the 18S rDNA data. Posterior probabilities are shown at nodes, followed by bootstrap support from the tree built for the same dataset with the ML method. Support values lower than 0.9 (BI) and 75 (ML) are not shown. Scale bar shows the substitution rate. The newly generated sequence is in bold. Different colors means different families.

**Figure 3 biology-13-00968-f003:**
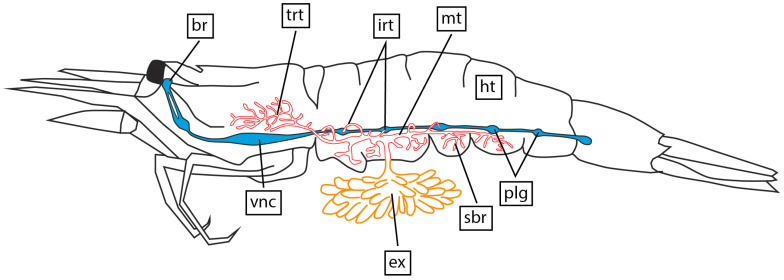
General scheme of *M. vancouverensis* female in the shrimp host. Red color—interna; yellow—externa; blue—host’s nervous system; br—brain of the host; ex—externa; irt—invasive rootlets; mt—main trunk; ht—host; plg—pleon ganglia; sbr—side branches; trt—trophic rootlets; vnc—ventral nerve cord.

**Figure 4 biology-13-00968-f004:**
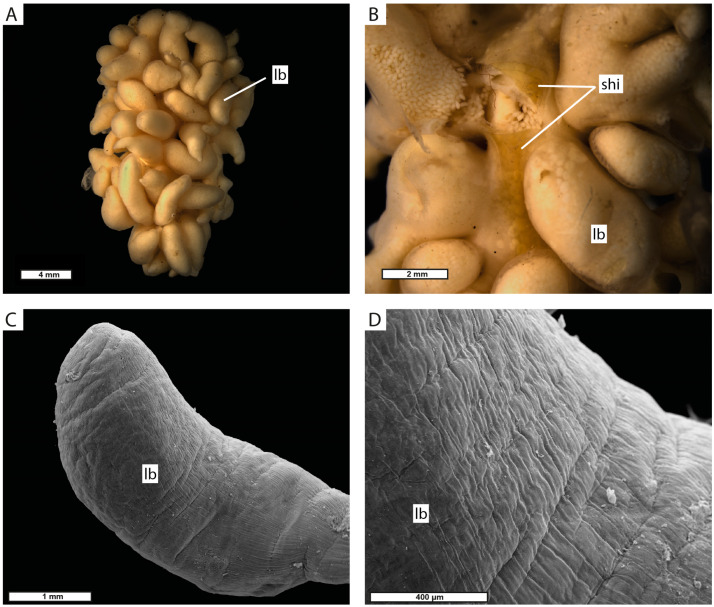
Externa of *M. vancouverensis*. (**A**) Whole externa “dorsal” view. (**B**) “Ventral” view to the central part of the interna. (**C**) General view of the externa lobe (SEM). (**D**) Fragment of externa lobe at higher magnification (SEM). lb, lobes of externa; shi, shield of the cuticle near the stalk.

**Figure 5 biology-13-00968-f005:**
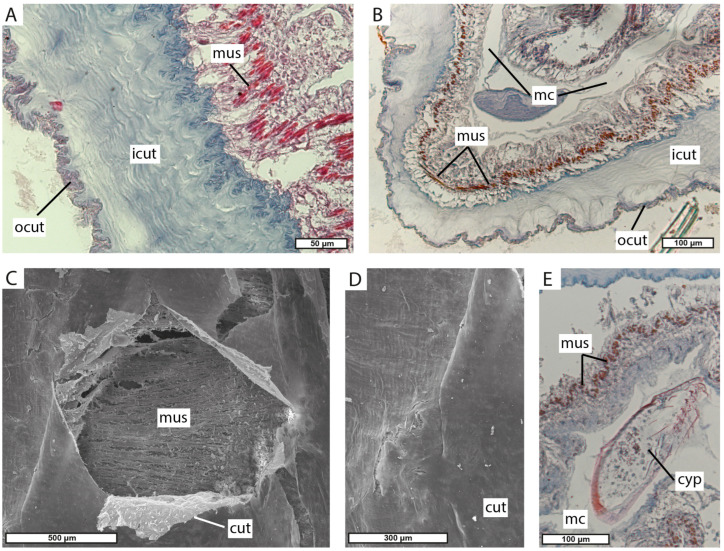
Morphology of externa lobes. (**A**,**B**) Histological transverse section of externa lobe. (**C**) Scanogram of the cuticle on the inner surface of the mantle cavity; in the center, the cuticle is damaged, and circular muscles are exposed. (**D**) Scanogram of the cuticle on the inner surface of the mantle cavity. (**E**) Histological section of cyprid larvae in the mantle cavity. cut, cuticle; cyp, cyprid larva; icut, the inner layer of cuticle; mc, mantle cavity; mus, muscular fibers; ocut, the outer layer of the cuticle.

**Figure 6 biology-13-00968-f006:**
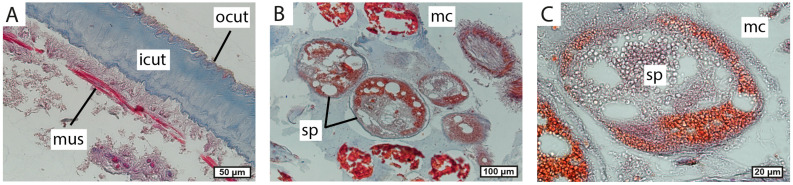
Morphology of the central part of externa. (**A**) Histological section of the body wall in the central part of externa. (**B**,**C**) Spermatogenic bodies in the mantle cavity. icut, the inner layer of cuticle; mc, mantle cavity; mus, muscular fibers; ocut, the outer layer of cuticle; sp, spermatogenic bodies.

**Figure 7 biology-13-00968-f007:**
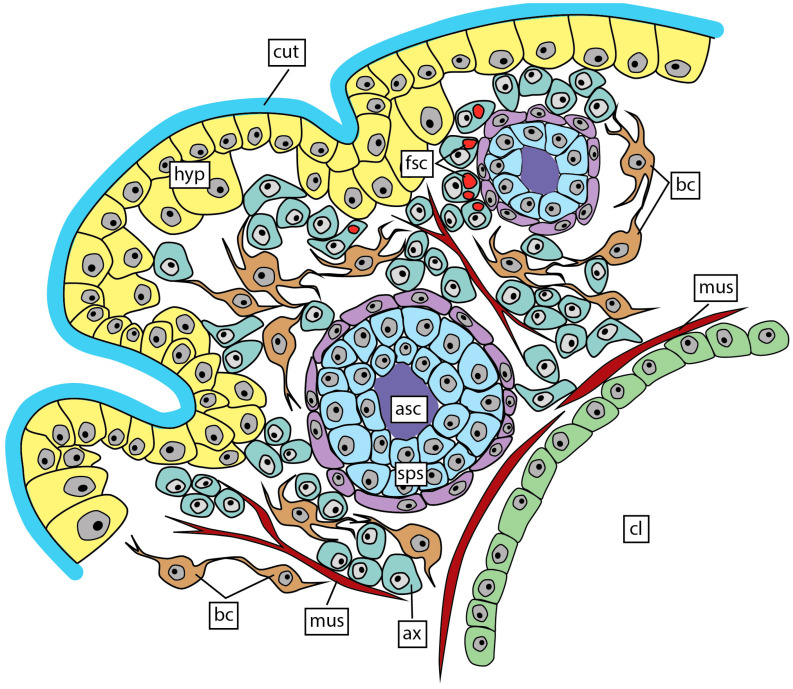
Scheme of the main trunk body wall. asc, aniline-stained content; ax, axial cells; cl, central lumen; bc, branched cells; cut, cuticle; fsc, cells with fuchsin-stained content; hyp, hypoderm; mus, muscular fibers; sps, spherical structure.

**Figure 8 biology-13-00968-f008:**
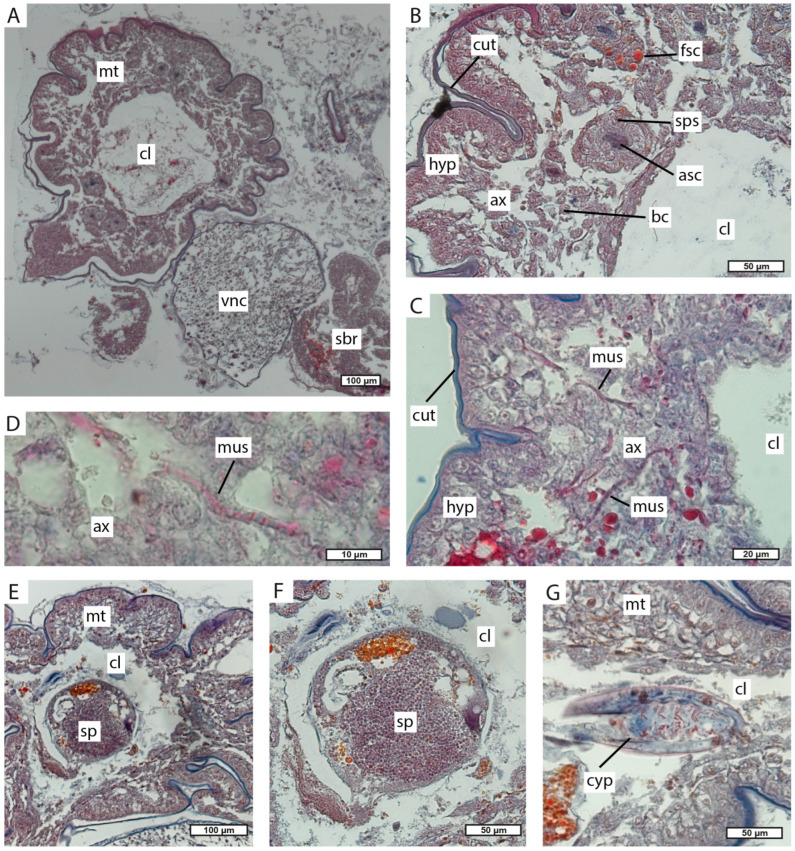
Histological sections of the main trunk. (**A**) Transverse section of the main trunk near the ventral nerve cord. (**B**–**D**) Transverse section of the body wall of the main trunk. (**E**,**F**) Spermatogenic bodies in the central lumen of the main trunk. (**G**) Transverse section of cyprid in the central lumen of the main trunk. asc, aniline-stained content; ax, axial cells; bc, branched cells; cl, central lumen; cyp, cyprid larva; fsc, cells with fuchsin-stained content; cut, cuticle; hyp, hypoderm; mt, main trunk; mus, muscular fibers; sbr, side branches; sp, spermatogenic bodies; sps, spherical structure; vnc, ventral nerve cord.

**Figure 9 biology-13-00968-f009:**
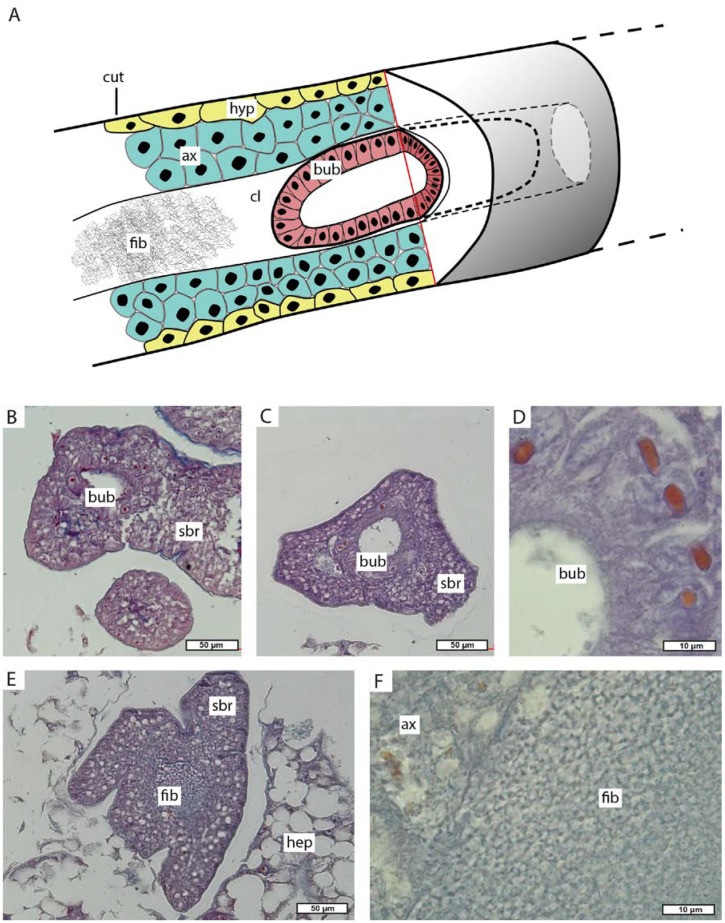
Morphology of the side branches of *M. vancouverensis* interna. (**A**) Scheme of side branch (trophic rootlet). (**B**) Histological transverse section of pleonal side branch (trophic rootlet). (**C**,**D**) Histological transverse section of pereonal side branch with a bubble-like structure (trophic rootlet). (**E**,**F**) Histological transverse section of pereonal side branch with fibrous content. ax, axial cells; bub, bubble-like structures; cl, central lumen; cut, cuticle; fib, fibrous content; hyp, hypoderm; hep, hepatopancreas; sbr, side branch.

**Figure 10 biology-13-00968-f010:**
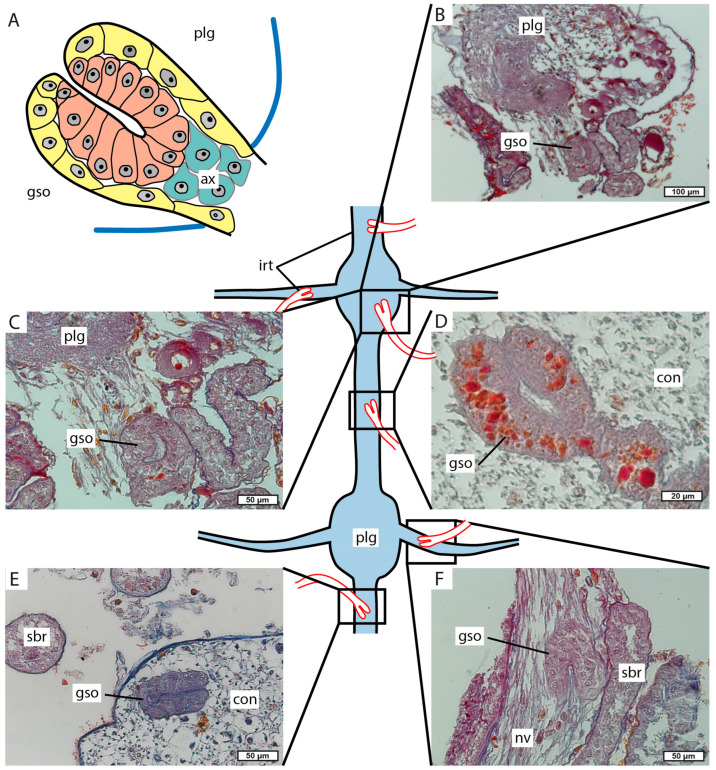
Invasive rootlets of *M. vancouverensis* interna. (**A**) Scheme of the goblet-shaped organs. (**B**,**C**) Goblet-shaped organs in pleonal ganglions. (**D**,**E**) Goblet-shaped organs in connectives. (**F**) Goblet-shaped organs in the segmental nerve. ax, axial cell layer; con, connectives; gso, goblet-shaped organs; irt, invasive rootlets; nv, segmental nerve; sbr, side branch (trophic rootlet); plg, pleonal ganglion.

**Table 1 biology-13-00968-t001:** List of taxa included in the analysis of 18S rDNA.

Species	Family	Reference	GenBank Accession Number
*Briarosaccus auratum* Noever, Olson, and Glenner, 2016	Peltogastridae	[24]	MN650344
*Briarosaccus regalis* Noever, Olson, and Glenner, 2016	Peltogastridae	[33]	LC742972
*Briarosaccus hoegi* Korn, Golubinskaya, Sharina, Noever, and Glenner, 2024	Peltogastridae	[34]	OR469043
*Cyphosaccus norvegicus* Boschma, 1962	Peltogasterellidae	[24]	MN650337
*Galatheascus striatus* Boschma, 1929	Peltogastridae	[24]	MN650343
*Heterosaccus californicus* Boschma, 1933	Sacculinidae	[35]	AY520657
*Heterosaccus dollfusi* Boschma, 1960	Sacculinidae	[36]	EU082413
*Heterosaccus lunatus* Phillips, 1978	Sacculinidae	[36]	EU082414
*Lernaeodiscus ingolfi* Boschma, 1928	Peltogastridae	[37]	MN625167
*Lernaeodiscus porcellanae* Müller, 1862	Peltogastridae	[37]	MN625166
*Lernaeodiscus rybakovi* Korn, Golubinskaya, Rees, Glenner, and Høeg, 2020	Peltogastridae	[37]	MN625165
*Loxothylacus panopaei* (Gissler, 1884)	Sacculinidae	[38]	AY265364
*Loxothylacus texanus* Boschma, 1933	Sacculinidae	[39]	L26517
*Mycetomorpha abyssalis* Kakui, 2024	Mycetomorphidae	[19]	LC799152
*Mycetomorpha vancouverensis* Potts, 1912	Mycetomorphidae	[23]	MH974514
*Mycetomorpha vancouverensis* Potts, 1912	Mycetomorphidae	This study	PQ408617
*Parasacculina leptodiae* (Guérin-Ganivet, 1911)	Polyascidae	[38]	AY265365
*Parasacculina oblonga* (Lützen and Yamaguchi, 1999)	Polyascidae	[38]	AY265367
*Parasacculina sinensis* (Boschma, 1933)	Polyascidae	[38]	AY265360
*Parasacculina yatsui* (Boschma, 1936)	Polyascidae	[26]	MG604305
*Paratriangulus galatheae* (Smith, 1906)	Peltogastridae	[24]	MN650340
*Parthenopea subterranea* Kossmann, 1874	Parthenopeidae	[40]	DQ826566
*Peltogaster curvata* Kossmann, 1874	Peltogastridae	[24]	MN650342
*Peltogaster lineata* Shiino, 1943	Peltogastridae	[34]	OR469045
*Peltogaster lineata* Shiino, 1943	Peltogastridae	[34]	OR469046
*Peltogaster paguri* Rathke, 1842	Peltogastridae	[40]	DQ826570
*Peltogaster paguri* Rathke, 1842	Peltogastridae	[36]	EU082415
*Peltogasterella sulcata* (Lilljeborg, 1859)	Peltogasterellidae	[40]	DQ826572
*Polyascus gregarius* (Okada and Miyashita, 1935)	Polyascidae	[38]	AY265363
*Polyascus planus* (Boschma, 1933)	Polyascidae	[38]	AY265368
*Polyascus polygeneus* (Lützen and Takahashi, 1997)	Polyascidae	[38]	AY265362
*Polyascus* sp.	Polyascidae	[41]	KM087478
*Ptychascus barnwelli* Andersen, Bohn, Høeg, and Jensen, 1990	Sacculinidae	[42]	MN523211
*Sacculina bicuspidata* Boschma, 1931	Polyascidae	[42]	KF561260
*Sacculina carcini* Thompson, 1836	Sacculinidae	[35]	AY520656
*Sacculina confragosa* Boschma, 1933	Polyascidae	[38]	AY265361
*Sacculina upogebiae* Shiino, 1943	Sacculinidae	[43]	KF539758
*Septosaccus rodriguezii* (Fraisse, 1878)	Peltogastridae	[40]	DQ826571
*Sesarmaxenos gedehensis* Feuerborn, 1932	Sacculinidae	[42]	KF561254
*Tortugaster boschmai* (Brinkman, 1936)	Peltogastridae	[24]	MN650341
*Triangulus munidae* Smith, 1906	Triangulidae	[24]	MN650339
**Outgroups**:			
*Ibla quadrivalvis* (Cuvier, 1817)		[35]	AY520655
*Poecilasma inaequilaterale* Pilsbry, 1907		[35]	AY520654
*Semibalanus balanoides* (Linnaeus, 1767)		[35]	AY520626

## Data Availability

Data are contained within the article.

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
