# Peer review of "Internal Morphology and Phylogenetic Position of Mycetomorpha vancouverensis (Pancrustacea: Rhizocephala), an Enigmatic Parasitic Barnacle"

_biology, 2024, doi:10.3390/biology13120968_

Round 1

Reviewer 1 Report

Comments and Suggestions for Authors

Reviewers' Comments to Authors:

The manuscript entitled Internal morphology and phylogenetic position of Mycetomorpha vancouverensis (Pancrustacea: Rhizocephala), an enigmatic parasitic barnacle”  by Miroliubov et al. demonstrates the scientific attempts to explain the evolution study of a parasitic barnacle in a wild-caught shrimp in Russian seawater areas. Using molecular and morphological analysis, the authors could newly find a clearer relationship between the target parasite and their counterpart clad.

Based on scientific consideration, the manuscript contains interesting findings that contribute to knowledge of parasitology and evolutionary information about a parasitic barnacle. However, it does contain some errors and unclear points that the authors must pay more attention to address and improve the quality of this manuscript to meet an acceptable high-standard journal. Additionally, there are many format errors throughout the manuscript, especially reference patterns that the authors should carefully correct.  The following minor and major concerns have been left for the authors to improve the quality of the current research work.

Title

Please italicize “Mycetomorpha vancouverensis” and any other genus and species terms throughout the manuscript.

Author name’s list.

Please correct and harmonize the font type in this section.

Simple Summary

Italicize “18S rDNA” and other terms to specifically indicate gene status.

Keywords

Please reconsider replacing the following uninformative keywords with proper ones; “interna and nervous system”.

1. Introduction

1) Lines 66-67. Please correct “(Lianguzova et al., 2023).”. The manuscript contains many errors in the reference format. Please pay more attention to improving these errors since they will affect the series of reference numbers in the whole manuscript.

2) To increase the flow, significant impacts caused by the target parasitic barnacle should be described in this part.

2. Materials and Methods

2.2 Histology

Line 117. Please correct the following unclear sentence; “The shrimps were dissected using a stereomicroscope…”. How can a stereomicroscope be used to dissect shrimp samples?

2.4 Molecular analysis

1) In PCR analysis, the authors should declare the quality and quantity of samples' DNA and properly narrate the protocols used to measure these parameters.

2) The authors should not ignore the protocols used to purify PCR products before being sequenced.

4) Line 137. Correct “mL-1”

5) 151. The concentration of “EtBr” should be indicated.

Results

1) In this part, the authors should initially narrate a number of samples harvested and used to study. Furthermore, clinical signs of the infected host shrimp with proper infected shrimp pictures. The sex ratio and size of infected shrimp should also be indicated. This information should be discussed and summarized properly.

2) 3.1 Molecular analysis

In this part, the exact number of samples and the number of DNA samples analyzed should be declared. In addition to this information, only one sequence was analyzed in this study, and the homological analysis of Blast searching in the GenBank database should be demonstrated.   

3) In Figure 2. It was unclear that parasitic organisms in Peltogastridae were distributed and mixed complicatedly with Peltogasterellidae and Polyascidae. Therefore, this critical point should be energetically discussed in the “Discussion” section.

References

Please carefully correct inconsistent formats and other errors in all references. There are many errors in this part, and it is not neat. Please take a look at the correct pattern in the journal guidelines. Additionally, errors of scientific name are always found throughout.

Author Response

We are grateful to the editor and the reviewers for their helpful comments. We tried to improve our manuscript according to your advice. 

The manuscript entitled “Internal morphology and phylogenetic position of Mycetomorpha vancouverensis (Pancrustacea: Rhizocephala), an enigmatic parasitic barnacle”  by Miroliubov et al. demonstrates the scientific attempts to explain the evolution study of a parasitic barnacle in a wild-caught shrimp in Russian seawater areas. Using molecular and morphological analysis, the authors could newly find a clearer relationship between the target parasite and their counterpart clad.

Based on scientific consideration, the manuscript contains interesting findings that contribute to knowledge of parasitology and evolutionary information about a parasitic barnacle. However, it does contain some errors and unclear points that the authors must pay more attention to address and improve the quality of this manuscript to meet an acceptable high-standard journal. Additionally, there are many format errors throughout the manuscript, especially reference patterns that the authors should carefully correct.  The following minor and major concerns have been left for the authors to improve the quality of the current research work.

Title

Please italicize “Mycetomorpha vancouverensis” and any other genus and species terms throughout the manuscript.

Done

Author name’s list.

Please correct and harmonize the font type in this section.

Done

Simple Summary

Italicize “18S rDNA” and other terms to specifically indicate gene status.

Despite rDNA being so widely used for the phylogenetic studies, there is surprisingly no uniformity in the way it must be written. Some authors state that the sequence coding small ribosomal subunit must be referred to as 18S rRNA gene, while others think the 18S rDNA is better, and some say just 18S; some people italicize it, others don’t. When choosing how to name this fragment, we took the recent articles from the “Biology” journal as an example, and 18S was not italicized there.

Keywords

Please reconsider replacing the following uninformative keywords with proper ones; “interna and nervous system”.

Done

1. Introduction

1) Lines 66-67. Please correct “(Lianguzova et al., 2023).”. The manuscript contains many errors in the reference format. Please pay more attention to improving these errors since they will affect the series of reference numbers in the whole manuscript.

Done

2) To increase the flow, significant impacts caused by the target parasitic barnacle should be described in this part.

Thank you for your comment, we have added this information. “Parasitic barnacles alter the metabolism, morphology and even behavior of infected specimens. The most striking example is the feminization of male crabs”

2. Materials and Methods

2.2 Histology

Line 117. Please correct the following unclear sentence; “The shrimps were dissected using a stereomicroscope…”. How can a stereomicroscope be used to dissect shrimp samples?

Thank you for your comment. We have corrected to “The shrimps were dissected in water under using a stereomicroscope  MBS 10 LOMO (Russia) to isolate parts of the interna and host’s ventral nerve cord.”

2.4 Molecular analysis

1) In PCR analysis, the authors should declare the quality and quantity of samples' DNA and properly narrate the protocols used to measure these parameters.

The fact that the quantity of DNA after PCR was enough for sequencing was estimated by visualisation through electrophoresis on agarose gel. The quality was estimated through analysis of chromatograms. 98,8% of the consensus sequence comes from overlap of two or more chromatograms. Aligned chromatograms were then checked by eye in Geneious Prime software. As an attachment to the reply, we provide an alignment of chromatograms used to make a consensus sequence, so the reviewer could check its quality.

2) The authors should not ignore the protocols used to purify PCR products before being sequenced.

The products of PCR were passed for sequencing without purification. As the resulting chromatograms were of well quality, purification wasn't necessary.

4) Line 137. Correct “mL-1”

It was intentionally μL, we had primer concentration 10 pmol in 1 μL. It is probably “-1” that was confusing, so now we changed it with “10 pmol/μL”.

5) 151. The concentration of “EtBr” should be indicated.

 The concentration was 0.5 μg/ml, we added this information.

Results

1) In this part, the authors should initially narrate a number of samples harvested and used to study. Furthermore, clinical signs of the infected host shrimp with proper infected shrimp pictures. The sex ratio and size of infected shrimp should also be indicated. This information should be discussed and summarized properly.

We have added the information about the total number of shrimps to the “material and methods” and added the photo of one infected specimen to Figure 1. However, due to the sampling method was not quantitative we are unable to make any conclusions about the populations and the infection rate.

2) 3.1 Molecular analysis

In this part, the exact number of samples and the number of DNA samples analyzed should be declared. In addition to this information, only one sequence was analyzed in this study, and the homological analysis of Blast searching in the GenBank database should be demonstrated.   

This information was added, we used one specimen of parasite to obtain DNA sequence. Text on the results of BLAST search was added.

3) In Figure 2. It was unclear that parasitic organisms in Peltogastridae were distributed and mixed complicatedly with Peltogasterellidae and Polyascidae. Therefore, this critical point should be energetically discussed in the “Discussion” section.

In the Results section we write: “In both BI and ML analyses, the family Peltogastridae appeared polyphyletic.” and “The Peltogasterellidae were also resolved as polyphyletic in our trees”. We expanded discussion on these results.

References

Please carefully correct inconsistent formats and other errors in all references. There are many errors in this part, and it is not neat. Please take a look at the correct pattern in the journal guidelines. Additionally, errors of scientific name are always found throughout.

Thank you for your comment we have improved this part.

Reviewer 2 Report

Comments and Suggestions for Authors

The manuscript biology-3298251 by Miroliubov et al. aims to study morphological characterization and to evaluate phylogenetic placement of parasitic barnacle Mycetomorpha vancouverensis collected from the South Kuril Island, among other rhizocephalan species. My major concern is that the phylogenetic position of Mycetomorpha vancouverensis among other rhizocephalan species has already been estimated by previous study (Høeg et al., 2019. N. J. Smith et al. (eds.), Parasitic Crustacea, Zoological Monographs 3). Thus, from the point of view phylogenetic relationships of this species in rhizocephalan species, I do not see new findings from this study. As for another flaw, the authors of this study did not include the 18S rDNA sequence of M. vancouverensis from Høeg et al. (2019) in their phylogenetic analysis. A direct comparison for the phylogenetic placement of M. vancouverensis individuals from this and previous study cannot be observed. Moreover, only single individual was sequenced for 18S rDNA in this study. Altogether, I am not sure if the content of this manuscript is worth a full contribution in this journal. I provide my comments below and hope the authors find them useful for improving the manuscript:

1.     Throughout the manuscript, genus and species names were not in italics. Please revise them all.

2.     Keywords: too many words; please reduce to 5-6 keywords. Avoid list words, which already appear in the title.

3.     Introduction, lines 71-74: this paragraph is too short without citing any references. Please expand it with relevant information and citation or just merge it with another paragraph.

4.     Introduction, lines 83-89: What methods were used for inferring phylogenetic relationships of the family Mycetomorphidae from previous studies? I think it is important to mention this point in this section, whether previous studies used molecular and/or morphological analyses in relation to phylogenetic analysis used in this study using 18S rDNA sequences.

5.     Introduction, lines 92-94: I do not think that the authors intended to conduct phylogenetic reconstruction of the family Mycetomorphidae based on the morphological structure of the interna and the invasive rootlets. This statement is somewhat confusing, in my opinion.

6.     Introduction, line 98: Did the authors intend to say this study was conducted to elucidate the phylogenetic position of M. vancouverensis among other rhizocephalans species? This study generated only one new sequence of 18S rDNA for M. vancouverensis. Thus, the statement in line 98 does not make sense.

7.     Materials and Methods, Sampling, lines 101-103: how many specimens were successfully collected? The authors mentioned the specimens were collected from three different localities?

8.     Molecular analysis: The author should incorporate 18S rDNA sequence of M. vancouverensis from Høeg et al. (2019) in this study. Then the authors can make direct comparison on the phylogenetic placement of M. vancouverensis for the sequences generated from this and previous study.

9.     Molecular analysis, line 155: Mycetomorpha à M. vancouverensis

10. Results: to make it consistent with the Materials and Methods section, I think it is better to put the results of morphological analyses before the results of molecular analysis

11. Results, molecular analysis, lines 180-183: this paragraph consist only one sentence. Please merge it with a subsequent paragraph or expand it with relevant information.

12. Discussion, Molecular analysis, lines 342-346: Høeg et al. (2019) in their article already conducted phylogenetic reconstruction using multiple loci of 16S, 18S and 28S gene sequences, didn’t they?

Author Response

The manuscript biology-3298251 by Miroliubov et al. aims to study morphological characterization and to evaluate phylogenetic placement of parasitic barnacle Mycetomorpha vancouverensis collected from the South Kuril Island, among other rhizocephalan species. My major concern is that the phylogenetic position of Mycetomorpha vancouverensis among other rhizocephalan species has already been estimated by previous study (Høeg et al., 2019. N. J. Smith et al. (eds.), Parasitic Crustacea, Zoological Monographs 3). Thus, from the point of view phylogenetic relationships of this species in rhizocephalan species, I do not see new findings from this study. As for another flaw, the authors of this study did not include the 18S rDNA sequence of M. vancouverensis from Høeg et al. (2019) in their phylogenetic analysis. A direct comparison for the phylogenetic placement of M. vancouverensis individuals from this and previous study cannot be observed. Moreover, only single individual was sequenced for 18S rDNA in this study. Altogether, I am not sure if the content of this manuscript is worth a full contribution in this journal. I provide my comments below and hope the authors find them useful for improving the manuscript:

Mycetomorpha was previously included into the molecular analysis once, in the publication by Høeg et al. (2019). The advantage of the previous analysis was including three loci: almost complete 18S, and short fragments of 28S rDNA and mitochondrial 16S rDNA. At the same time, using three loci delimits the number of species that can be taken into the analysis. There are no 16S or/and 28S data on many genera within the Peltogastridae, like Tortugaster, Galatheascus, Septosaccus, Paratriangulus, as well as on the other “kentrogonid” rhizocephalans. In the publication by Høeg et al. (2019) the main point on the Mycetomorpha phylogeny was transferring it from “akentrogonids” to “kentrogonids”, and specifically close to Peltogastridae.

In the present analysis, we aimed to figure out the phylogenetic position of Mycetomorpha more precisely. For this, more various representatives of “kentrogonids” were taken into the analysis. Together with that, we removed “akentrogonids” for two reasons. First, they had been shown to be distant from Mycetomorpha (Høeg et al., 2019; our preliminary analyses). Second, they form very long branches, and this can strongly affect the resulting tree topology. Thus, taxa sampling in our analysis differs substantially from the one of Høeg et al. (2019). The results are also different: Mycetomorpha was shown to be close not to all peltogastrids, but to a specific group of genera. Moreover, peltogastrids were resolved polyphyletic, unlike in Høeg et al. (2019).

In the revised manuscript we expanded the explanation how our analysis differs from the previous and why it was necessary.

Regarding the sequence MH974514 from Høeg et al. (2019), in the Results section we firstly compare that one with the newly obtained, and they are identical in the overlapping part. We do not add this sequence into the analysis because adding identical sequences may negatively affect the tree. We use our sequence for the phylogeny reconstruction because it is a little bit longer.

  1.     Throughout the manuscript, genus and species names were not in italics. Please revise them all.

Done

  1.     Keywords: too many words; please reduce to 5-6 keywords. Avoid list words, which already appear in the title.

Done

  1.     Introduction, lines 71-74: this paragraph is too short without citing any references. Please expand it with relevant information and citation or just merge it with another paragraph.

We have added citation and merged with the previous paragraph

  1.     Introduction, lines 83-89: What methods were used for inferring phylogenetic relationships of the family Mycetomorphidae from previous studies? I think it is important to mention this point in this section, whether previous studies used molecular and/or morphological analyses in relation to phylogenetic analysis used in this study using 18S rDNA sequences.

Thank you for the comment we have added this information to this paragraph and rewritten it.

  1.     Introduction, lines 92-94: I do not think that the authors intended to conduct phylogenetic reconstruction of the family Mycetomorphidae based on the morphological structure of the interna and the invasive rootlets. This statement is somewhat confusing, in my opinion.

We changed and expanded the explanation on aims and approaches of this study in this paragraph. 

  1.     Introduction, line 98: Did the authors intend to say this study was conducted to elucidate the phylogenetic position of M. vancouverensis among other rhizocephalans species? This study generated only one new sequence of 18S rDNA for M. vancouverensis. Thus, the statement in line 98 does not make sense.

Thank you for your comment. First, sequence of 18S rDNA was used to ensure morphological identification of M. vancouverensis; according to the data on other parasitic barnacles, lack of difference in 18S is enough to assume conspecificity.

Second, we used our sequence within a dataset different from the one in the previous molecular study on Mycetomorpha (Høeg et al., 2019). It comprised 18S sequences from many “kentrogonid” species, and excluded “akentrogonids” (explained in detail above). That is why we have results partially different from those of Høeg et al. (2019): Mycetomorpha is resolved related to certain peltogastrids, but the peltogastrids themselves appear polyphyletic.

  1.     Materials and Methods, Sampling, lines 101-103: how many specimens were successfully collected? The authors mentioned the specimens were collected from three different localities?

Thank you for your comment, we made the text more clear in this part. 

From these three localities, three infected shrimps were collected in total.

  1.     Molecular analysis: The author should incorporate 18S rDNA sequence of M. vancouverensis from Høeg et al. (2019) in this study. Then the authors can make direct comparison on the phylogenetic placement of M. vancouverensis for the sequences generated from this and previous study.

This sequence is incorporated in the analysis, however in the Table 1 it was missing due to our mistake; it was added to Table 1 in the revised manuscript. In the first paragraph of the results, we compare our sequence with the one from Høeg et al. (2019), and state that they are 100% identical in the overlapping part. In the revised manuscript, we provide additional information from BLAST on the identities with M. abyssalis and other rhizocephalans.

We do not add the sequence MH974514 from Høeg et al. (2019) into the alignment used for BI and ML analyses. That was done for a reason: adding identical sequences may negatively affect the tree. We used our sequence for the phylogeny reconstruction because it is a little bit longer than that of Høeg et al. (2019).

  1.     Molecular analysis, line 155: Mycetomorpha à M. vancouverensis

That was corrected.

  1. Results: to make it consistent with the Materials and Methods section, I think it is better to put the results of morphological analyses before the results of molecular analysis

Thank you for your comment, but in order to follow the flow of our manuscript we changed the order in the “material and methods”

  1. Results, molecular analysis, lines 180-183: this paragraph consist only one sentence. Please merge it with a subsequent paragraph or expand it with relevant information.

We expanded this paragraph by adding more information on BLAST results.

  1. Discussion, Molecular analysis, lines 342-346: Høeg et al. (2019) in their article already conducted phylogenetic reconstruction using multiple loci of 16S, 18S and 28S gene sequences, didn’t they?

That is indeed so, but the analyses from Høeg et al. (2019) and from the present study are different. The advantage of the previous analysis was including three loci. At the same time, using three loci delimits the number of species that can be taken into the analysis. There are no 16S or/and 28S data on many genera within the Peltogastridae, like Tortugaster, Galatheascus, Septosaccus, Paratriangulus, as well as on the other “kentrogonid” rhizocephalans. In the publication by Høeg et al. (2019) the main point on the Mycetomorpha phylogeny was transferring it from “akentrogonids” to “kentrogonids”, and specifically close to Peltogastridae.

In the present analysis, we aimed to figure out the phylogenetic position of Mycetomorpha more precisely. For this, more various representatives of “kentrogonids” were taken into the analysis. Together with that, we removed “akentrogonids”: they had been shown to be distant from Mycetomorpha, and they form very long branches, and this can strongly affect the resulting tree topology.

Thus, taxa sampling in our analysis differs substantially from the one of Høeg et al. (2019). The results are also different: Mycetomorpha was shown to be close not to all peltogastrids, but to certain specific genera. Moreover, peltogastrids were resolved polyphyletic, unlike in Høeg et al. (2019).

Round 2

Reviewer 1 Report

Comments and Suggestions for Authors

Reviewers' Comments to Authors:

The revised manuscript entitled Internal morphology and phylogenetic position of Mycetomorpha vancouverensis (Pancrustacea: Rhizocephala), an enigmatic parasitic barnacle”  by Miroliubov et al. demonstrates the scientific attempts to explain the evolution study of a parasitic barnacle in a wild-caught shrimp in Russian seawater areas. Using molecular and morphological analysis, the authors could newly find a clearer relationship between the target parasite and their counterpart clad.

In the revised version, the authors have substantially responded to almost all comments raised previously. The “accept” recommendations will be suggested after correcting the following minor issues.  

Line 68. Please rearrange the following reference numbers from “[15,16, 11-14]” to “[11-14, 15,16,]” and the other places throughout the manuscript.

Line 106. The reference “(Høeg et al., 2020)” must be replaced with the proper reference number.

Line 161. Correct “Molecular analysis” to “2.2 Molecular analysis”.

Line 180. Correct the following unclear content from Sequencing was performed with PCR primers on an ABI Prism 3500xl genetic analyser (Applied Biosystems, MA, USA).” to “Sequencing of the target PCR products, was further performed with PCR primers on an ABI Prism 3500xl genetic analyser (Applied Biosystems, MA, USA).”  

Line 239. Unitalicized “spp.”.

Author Response

We are grateful to the editor and the reviewers for their helpful comments. We tried to improve our manuscript according to your advice. 

The revised manuscript entitled “Internal morphology and phylogenetic position of Mycetomorpha vancouverensis (Pancrustacea: Rhizocephala), an enigmatic parasitic barnacle”  by Miroliubov et al. demonstrates the scientific attempts to explain the evolution study of a parasitic barnacle in a wild-caught shrimp in Russian seawater areas. Using molecular and morphological analysis, the authors could newly find a clearer relationship between the target parasite and their counterpart clad.

In the revised version, the authors have substantially responded to almost all comments raised previously. The “accept” recommendations will be suggested after correcting the following minor issues.  

Line 68. Please rearrange the following reference numbers from “[15,16, 11-14]” to “[11-14, 15,16,]” and the other places throughout the manuscript.

Done

Line 106. The reference “(Høeg et al., 2020)” must be replaced with the proper reference number. 

Done

Line 161. Correct “Molecular analysis” to “2.2 Molecular analysis”.

Done

Line 180. Correct the following unclear content from “Sequencing was performed with PCR primers on an ABI Prism 3500xl genetic analyser (Applied Biosystems, MA, USA).” to “Sequencing of the target PCR products, was further performed with PCR primers on an ABI Prism 3500xl genetic analyser (Applied Biosystems, MA, USA).”  

Done 

Line 239. Unitalicized “spp.”.

Done

Reviewer 2 Report

Comments and Suggestions for Authors

I have read the revised version of the manuscript. I am pleased to note that most of my major concerns in the previous round of revisions have been addressed by the authors. I have small minor corrections in the new version. After correction, I recommend the manuscript for indexing.

line 273: (Fig 4 A) change to (Figure 4 A)

line 275: (Fig 4 B) change to (Figure 4 B)

line 276: (Fig 4 C, D) change to (Figure 4 C, D)

Figure 7 should be mentioned in the main text

Author Response

We are grateful to the editor and the reviewers for their helpful comments. We tried to improve our manuscript according to your advice. 

I have read the revised version of the manuscript. I am pleased to note that most of my major concerns in the previous round of revisions have been addressed by the authors. I have small minor corrections in the new version. After correction, I recommend the manuscript for indexing.

line 273: (Fig 4 A) change to (Figure 4 A)

Done

line 275: (Fig 4 B) change to (Figure 4 B)

Done

line 276: (Fig 4 C, D) change to (Figure 4 C, D)

Done

Figure 7 should be mentioned in the main text

Done